# Shiny-SoSV: A web-based performance calculator for somatic structural variant detection

**Tingting Gong**[1,2], **Vanessa M. Hayes**[1,2,3], **Eva K. F. Chan**[1,3]*

**1** Garvan Institute of Medical Research, Darlinghurst, New South Wales, Australia, **2** Central Clinical School, University of Sydney, Camperdown, New South Wales, Australia, **3** St Vincent's Clinical School, University of New South Wales, Randwick, New South Wales, Australia

* eva.chan@health.nsw.gov.au

**Data Availability Statement:** All input data to the prediction models, R scripts for model fitting, and source code for the Rshiny app are available from the Git Hub repository: https://github.com/tgong1/Shiny-SoSV.

## Abstract

Somatic structural variants are an important contributor to cancer development and evolution. Accurate detection of these complex variants from whole genome sequencing data is influenced by a multitude of parameters. However, there are currently no tools for guiding study design nor are there applications that could predict the performance of somatic structural variant detection. To address this gap, we developed Shiny-SoSV, a user-friendly web-based calculator for determining the impact of common variables on the sensitivity, precision and F1 score of somatic structural variant detection, including choice of variant detection tool, sequencing depth of coverage, variant allele fraction, and variant breakpoint resolution. Using simulation studies, we determined singular and combinatoric effects of these variables, modelled the results using a generalised additive model, allowing structural variant detection performance to be predicted for any combination of predictors. Shiny-SoSV provides an interactive and visual platform for users to easily compare individual and combined impact of different parameters. It predicts the performance of a proposed study design, on somatic structural variant detection, prior to the commencement of benchwork. Shiny-SoSV is freely available at https://hcpcg.shinyapps.io/Shiny-SoSV with accompanying user's guide and example use-cases.

## Introduction

Somatic structural variants (SVs) are large ($> 50$ bp) genomic rearrangements that arise in tumours and could directly contribute to cancer development and progression [1–3]. The advent of next generation sequencing (NGS) has facilitated an increase in the efficiency and accuracy of detecting somatic variants in cancer genomes. However, due to limitations of short-read NGS, large SVs can only be inferred through alignment signatures. For example, read pairs that map farther or closer apart than expected are suggestive of a deletion or insertion event respectively, while read pairs aligned with inconsistent orientation are indicative of an inversion or duplication event. Attempts to improve SV detection have thus spurred many

**Funding:** This work was financially supported by
an Australian Government Research Training
Program Scholarship awarded to T.G., by a
Movember Revolutionary Team Award (PRoMis)
from Movember Australia and the Prostate Cancer
Foundation Australia, by the University of Sydney
Foundation and Petre Foundation funding received
by V.M.H. The funders had no role in study design,
data collection and analysis, decision to publish, or
preparation of the manuscript.

**Competing interests:** The authors have declared
that no competing interests exist.

computational developments, resulting in the publication and availability of numerous SV
detection tools (e.g. [4–8]). These tools are based on different, sometimes overlapping algo-
rithms and multiple evaluation studies have shown that these tools differ substantially in their
sensitivity and breakpoint precision [9–12].

SV detection is further complicated in cancer samples due to variable tumour purity (pro-
portion of cancer to non-cancerous cells in a sample) and variant allele frequency (VAF, pro-
portion of the sequencing reads captured as harbouring the variant at a given genomic locus).
Although histopathology can provide an overall estimate of tumour purity, it typically only
reflects an approximate estimate of VAF, due to further complexity created through tumour
sub-clonality and variability between histopathology and sequencing samples. Although
deeper sequencing may increase sensitivity [11], numerous studies have shown that the benefit
of increasing coverage does saturate (e.g. [13]). Thus, knowing how much sequencing depth to
increase is not obvious with decisions typically based on experience or "gut instinct".

Taken together, there are many alterable variables that can affect somatic SV detection in
NGS analyses. Yet, informed decisions on these variables are rarely easily made, especially by
non-bioinformaticians in clinical settings. To address this, we have developed Shiny-SoSV, a
web-based interactive application to help estimate the performance (sensitivity, precision and
F1 score) of somatic SV detection based on four user-modifiable parameters: VAF, sequencing
depth of tumour and matched-normal samples, and SV breakpoint precision.

## Materials and methods

### Simulation of structural variants

To evaluate the sensitivity and accuracy of SV detection, a simulation study was devised. Two
sets of SVs including 200 (somatic) and 2,000 (germline) of each of six SV types (deletion
(DEL), duplication (DUP), inversion (INV), domestic insertion (DINS), foreign insertion
(FINS) and translocation (TRA)) were simulated with SVEngine [14], as previously described
[10], totalling 1,200 somatic and 12,000 germline SVs (S1 Table). The numbers of SVs simu-
lated were based on previous studies that have suggested the presence of 5,000 to 10,000 poly-
morphic SVs in the human genome [15] and between a handful to up to 1000 somatic SVs
across a range of tumour types [16]. These findings support what we have also observed in our
own work [17, 18]. It is worth noting that, very similar results were observed when the number
of germline SVs were reduced to match the number of somatic SVs except for two callers (S1
File). For each SV type, variable lengths were included ranging from 50 bp to 1,000,000 bp.
Here, SV length is the length of the rearranged sequence; for example, the SV length of a TRA
is the length of the translocated piece of DNA fragment.

SVs were randomly distributed along the genome (GRCh38), masking N-gap, centromeric
and telomeric regions. N-gap and centromeric regions were based on gap [last updated 24/12/
2013] and centromeres [last update 17/08/2014] tables from the University of California at
Santa Cruz (UCSC) Genome Browser, while telomeric regions are defined as the 10 Mbp of
both ends of each chromosome of GRCh38.

### Simulation of short paired-end reads

NGS data was simulated to reflect the typical matched tumour-normal whole genome
sequencing (WGS) approach for cancer genomics, namely using paired-end short-read
sequencing, where genomic DNA is fragmented to a typical size range of 500 bp (insert size)
and the two ends of each fragment sequenced inwards to up to 150 bp (read length).

The 12,000 simulated germline SVs were spiked into both the normal and tumour genomes
(FASTA format), while the 1,200 simulated somatic SVs were additionally spiked into the

tumour genome. Paired short-reads were sampled using SVEngine [14] from the altered FASTA to 120x depth of coverage. Insert size of read-pairs was simulated with a normal distribution with mean of 500 bp and standard deviation of 100 bp, while read length was set to 150 bp. Other parameters were kept as SVEngine default, including random embedding of sequencing error and small variants (base error rate = 0.02, rate of mutation = 0.001, fraction of indels = 0.15, probability an indel is extended = 0.3). Sequencing reads (FASTQ format) were aligned to human genome reference GRCh38 using BWA-MEM v0.7.17-r1194 [19], generating alignment files (BAM format). Different depths of sequencing coverage were obtained by subsampling from the two 120x datasets using Picard DownsampleSam (http://picard.sourceforge.net). Different tumour purity levels (VAF) were emulated by merging varying ratios of normal to tumour aligned reads, using Picard MergeSamFiles, to create the final tumour files (BAM). Here, tumour purity and VAF are used interchangeably as SVs are assumed independent of each other.

The simulation was conducted to include a comprehensive combination of depths of coverage of the normal and tumour samples, VAF and SV breakpoint precision threshold (explained below in section "Defining true positive calls and concordant callsets"). The following parameter values were simulated:

- Normal sample coverage: 20x, 30x, 40x, 50x, 60x, 70x, 80x, 90x, 100x;

- Tumour sample coverage: 20x, 30x, 40x, 50x, 60x, 70x, 80x, 90x, 100x;

- VAF: 0.05, 0.10, 0.20, 0.30, 0.40, 0.50, 0.60, 0.70, 0.80, 0.90, 1.00;

- Breakpoint precision threshold (bp): 2, 10, 20, 40, 60, 80, 100, 120, 140, 160, 180, 200.

In all, this resulted in a total of 891 tumour/normal pairs of BAM files, encompassing all exhaustive combinations of parameter values.

## Somatic SV detection

Somatic SVs were detected using five SV callers, namely Manta [4], Lumpy [5], GRIDSS [6], Delly [7] and SvABA [8] for each tumour/normal pair and high-confidence calls were post-filtered as described previously [10]. Each SV caller was executed using default parameters.

These five callers were chosen to provide a wide representation of different SV detection methods and because they have been shown to be best performers within their class in recent benchmarking studies [9–12]. While, both Delly and Lumpy use discordant read-pair and split-read methods, Lumpy integrates the two methods into a single SV detection step, called "evidence clustering", whereas Delly uses them in separate calling and refining steps [10]. Manta, GRIDSS and SvABA further use local-assembly with different methods of targeted assembly, windowed local assembly and genome-wide break-end assembly respectively [10]. Additionally, Manta, Lumpy, GRIDSS and Delly were identified as popular tools among 46 callers published from 2009 to 2017, based on the criteria of Web of Science counts in a recent benchmarking study [12]. While SvABA was published in 2018 and not included in that study, it has shown good performance for both germline and somatic SV detection in other studies [9, 10].

## Defining true positive calls and concordant callsets

Candidate SV callsets were compared against "true" simulated SV sets as previously described [10]. In brief, a true positive (TP) SV call must meet two criteria: i) reported SV type must match the simulated type and ii) detected breakpoints must be within $T$ bp from the simulated breakpoints. To pass the second criterion, all breakpoint positions corresponding to a SV

event must be within a prescribed breakpoint precision threshold ($T$). We have applied varying values of $T$, as mentioned above.

The total number of true positives is calculated as the number of calls in a callset satisfying both TP criteria. The number of false positives (FP) is the number of SV calls in a callset not satisfying either TP criteria. Thus, following the confusion matrix, we define: sensitivity = TP / (TP + FN) and precision = TP / (TP + FP). F1 score is calculated as 2 * (sensitivity * precision) / (sensitivity + precision).

SV detection performance were also calculated for each SV type (DEL, DUP, INV, DINS, FINS and TRA) using the criteria as described above where possible, or with minor modifications as follows. For DINS and TRA, we require that both breakends (BNDs) of their fusion junctions to satisfy the TP criteria described to be considered as a true positive [10]. Precision was not calculated for DINS and FINS, but for a general "INS" as reported by the callers. As INS is not reported by Lumpy, precision estimate of INS is not available for this SV caller. Similarly, neither DUP nor INS are reported by SvABA, however, the presence of these two SV types can be inferred through post-analysis, albeit indistinguishable [10]. Therefore, the precision of DUP/INS was combined for SvABA. Furthermore, F1 scores of INS were calculated using sensitivity estimates for either DINS or FINS. Similarly, for SvABA, DUP/INS F1 scores were calculated using sensitivity values from DINS, FINS or DUP.

The union and intersection callsets for each pair of SV callers were also derived, resulting in another 40 callsets for each simulated dataset. Two SV calls were considered the same if they have matching SV type [10] and their reported breakpoint positions are within 5 bp of each other. The union callset of two SV callers is SV calls detected by either Caller_1 or Caller_2, while the intersection callset are calls reported by both Caller_1 and Caller_2. In both union and intersection cases, Caller_1 is the "dominant caller" such that any overlapping calls in the final callset is taken from the output from Caller_1 (including coordinates and SV types).

## Predictive model selection

A generalised additive model (GAM) was used to assess and predict the relationships between predictor variables (VAF, tumour coverage, normal coverage and breakpoint precision threshold $T$) and response variables (sensitivity and precision and F1 score) for each SV caller and all pairs of callers. The choice of GAM was motivated by non-linear relationships observed between some response and predictor variables.

In particular, we observed VAF to have a non-linear effect on sensitivity for all SV callers (Fig 1A and S1A Fig) and precision on some SV callers (Fig 2A). However, the non-linear effect of VAF on precision is less obvious at lower breakpoint precision resolution (higher $T$) and for some SV callers, such as GRIDSS (S1B Fig). Tumour coverage has a greater non-linear impact on sensitivity and precision on samples at low VAF (S2 Fig). Different SV callers have different breakpoint detection resolution, as previously described [10] and SvABA was found to have the lowest breakpoint resolution among the five SV callers examined (Fig 4). Consequently, breakpoint precision threshold ($T$) has greater effect for SvABA and less so for other callers (S3 Fig). Additionally, most SV callers have lower breakpoint resolution at low tumour coverage and low VAF (e.g. 20x and 0.2 respectively in Fig 4). The impact of $T$ on sensitivity, therefore, appears to be variably dependent on tumour coverage and VAF. For example, the sensitivity of Manta increases with increasing $T$ at low VAF (e.g. 0.2) and is only obvious at tumour coverage < 60x (S3A Fig). Similarly, precision is predominantly influenced by $T$ at low tumour coverage and VAF (S3B Fig). Normal coverage has no notable impact, except for Manta, which was observed to have only a minor effect on sensitivity and only at low normal coverage (< 40x) (S4 Fig). The observed relationships between F1 score and predictor variables (Fig 3) were similar as for sensitivity.

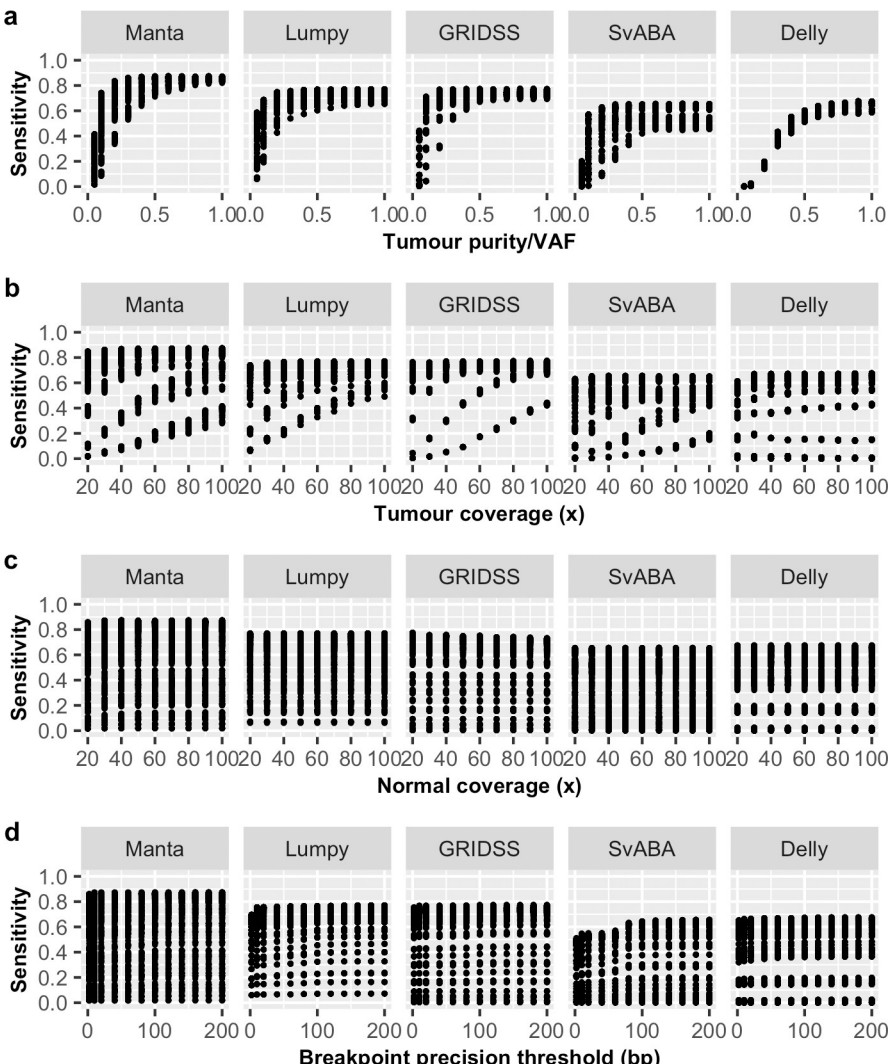

**Fig 1. Relationship between sensitivity and predictor variables.** Shown are the effects of (a) tumour purity/VAF, (b) tumour coverage, (c) normal coverage and (d) breakpoint precision threshold on somatic SV detection sensitivity for five SV callers (Manta, Lumpy, GRIDSS, SvABA, Delly).

In order to select the model with the most appropriate set of parameters and that best explain the relationships between predictor and response variables, we examined eight models (Model (1)–(8)) for sensitivity, precision and F1 score, independently for each SV caller. To account for non-linear joint relationships, smooth functions (*f*) were applied to the tensor product of selected predictor variables. Smoothing parameters were estimated using restricted maximum likelihood (REML), rather than the default generalised cross validation (GCV), to avoid local minima [20]. Further, as the response variables are proportions with values between 0 and 1, the beta regression (betar) with logistic link function was used.

$$Y \approx f_1(VAF \otimes Tumour\ coverage) + f_2(VAF \otimes T) + f_3(VAF \otimes Normal\ coverage) \\ + f_4(Tumour\ coverage \otimes T) \tag{1}$$

$$Y \approx f_1(VAF \otimes Tumour\ coverage) + f_2(VAF \otimes T) + f_3(VAF \otimes Normal\ coverage) \tag{2}$$

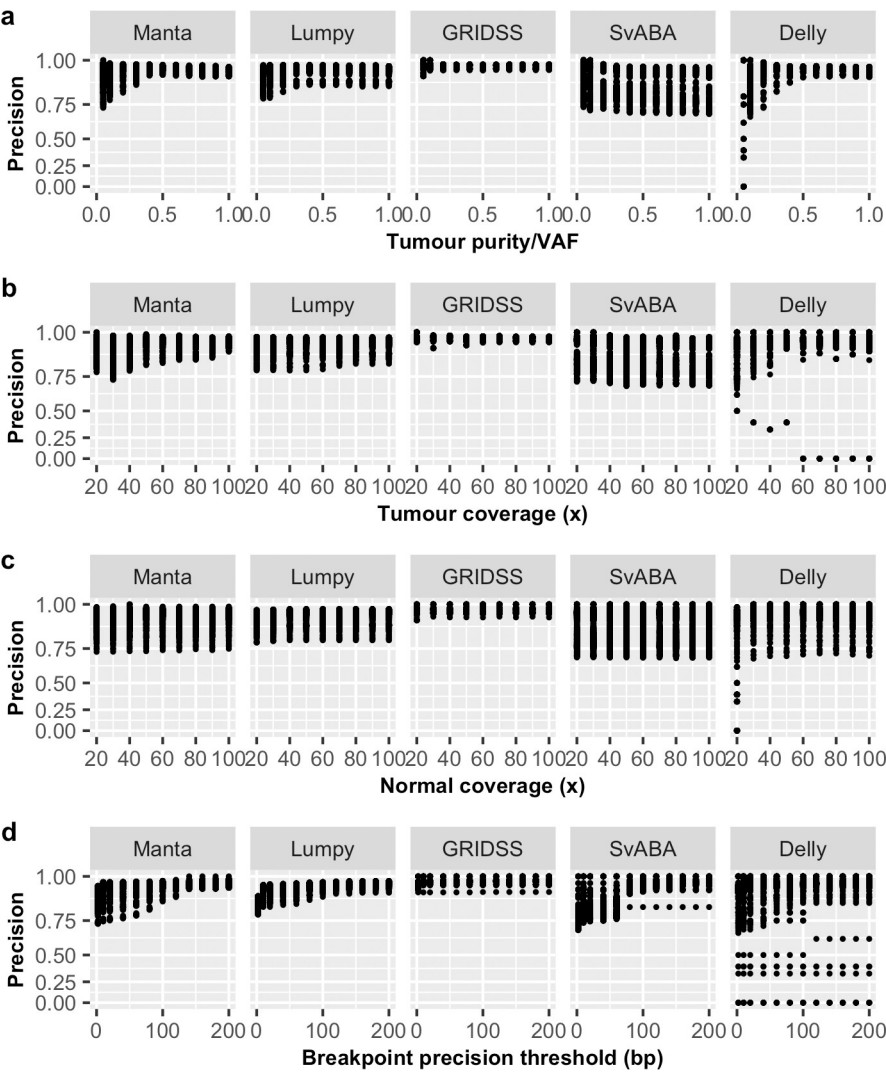

**Fig 2. Relationship between precision and predictor variables.** Shown are the effects of (a) tumour purity/VAF, (b) tumour coverage, (c) normal coverage and (d) breakpoint precision threshold on somatic SV detection precision, represented in log-scale, for five SV callers (Manta, Lumpy, GRIDSS, SvABA, Delly).

$$Y \approx f_1(VAF \otimes Tumour\ coverage) + f_2(VAF \otimes T) + f_3(Tumour\ coverage \otimes T) + Normal\ coverage \quad (3)$$

$$Y \approx f_1(VAF \otimes Tumour\ coverage) + f_2(VAF \otimes T) + Normal\ coverage \quad (4)$$

$$Y \approx f_1(VAF \otimes Tumour\ coverage) + f_2(Tumour\ coverage \otimes T) + Normal\ coverage \quad (5)$$

$$Y \approx f_1(VAF \otimes Tumour\ coverage) + T + Normal\ coverage \quad (6)$$

$$Y \approx f_1(VAF \otimes T) + Tumour\ coverage + Normal\ coverage \quad (7)$$

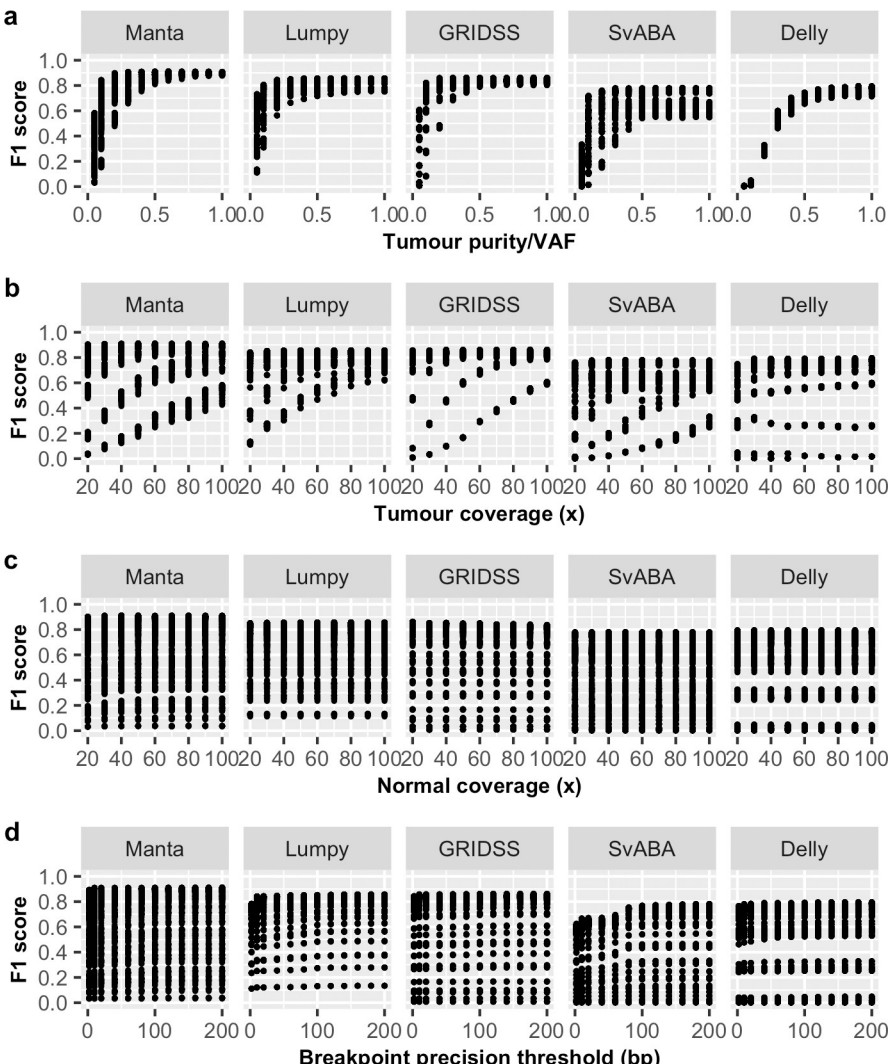

**Fig 3. Relationship between F1 score and predictor variables.** Shown are the effects of (a) tumour purity/VAF, (b) tumour coverage, (c) normal coverage and (d) breakpoint precision threshold on somatic SV detection F1 score for five SV callers (Manta, Lumpy, GRIDSS, SvABA, Delly).

$$Y \approx VAF + T + Tumour\ coverage + Normal\ coverage \qquad (8)$$

where Y is response variable, which is sensitivity, precision or F1 score.

To select the models for sensitivity, precision and F1 score, we used cross-validation for accuracy calculation of each model for each SV caller. Each time, one simulated value of a variable (e.g. all data with 20x tumour coverage) was left out for model fitting and error estimation. The average root mean squared error (RMSE) and mean absolute error (MAE) for each candidate model and five SV callers are reported in Table 1 and S2 Table respectively. If multiple models achieved similarly low RMSE and MAE, the simplest model was selected to avoid overfitting. As a result, for sensitivity, Model 2, 5 and 3 was selected for Manta, Lumpy and SvABA respectively, and Model 6 for GRIDSS and Delly. In addition, we found that matched normal coverage has no significant effect on Lumpy, SvABA and Delly sensitivity (S3 Table), therefore,

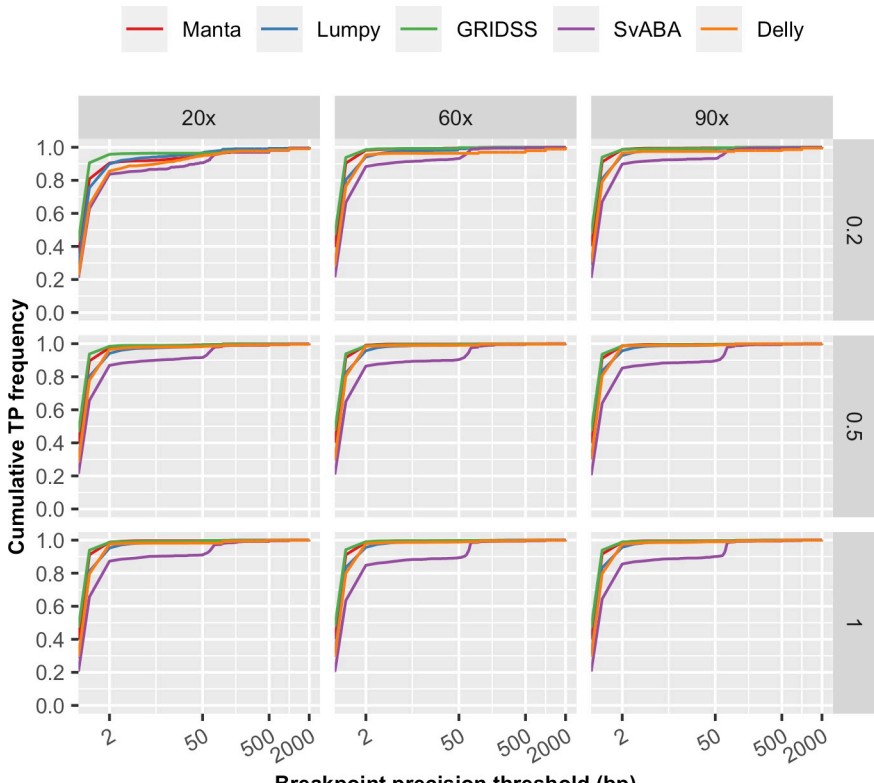

**Fig 4. Breakpoint resolution of structural variant calling.** Results are based on simulation SV Set 1 with tumour coverage of 20x, 60x and 90x, matched normal coverage of 60x and variant allele frequency of 0.2, 0.5 and 1. This figure shows the cumulated frequency of the number true positive structural variants detected by each caller for breakpoint precision thresholds less than 2,000 bp.

the term was excluded in the models for these SV callers. For precision, Model 3 was selected for Manta, Model 2 for Lumpy and Delly and Model 5 for GRIDSS and SvABA. For F1 score, Model 2, 6 and 3 was selected for Manta, GRIDSS and SvABA respectively, and Model 5 for Lumpy and Delly. In the chosen models for F1 score, matched normal coverage was not observed to have any significant effect on SvABA and Delly (S3 Table), and thus normal coverage was excluded from the final F1 score prediction models for these two callers.

Recognising performance differences for different SV types, we repeated the same evaluation analysis individually for each of the six SV types, where possible (S1 File), with the most appropriate prediction model identified using the same approach (S4 Table). Despite detection performance variability observed among SV types, it is noteworthy that the impact of predictor variables on the detection of individual SV type (S5–S16 Figs) is similar to the impact on overall performance (Figs 1–3).

The R package *mgcv* default of thin plate regression spline (tp) was used as the smoothing basis. As tp tend to result in lower mean squared error [20], it was implemented for our purpose. Basis dimension was set as the default for all smoothing terms.

The models for pairwise union and intersection of SV callers were chosen to be the same as the dominant caller. GAM models were then fitted for each SV caller and each pair of SV callers.

**Table 1. Predictive model comparison and selection.**

| Error Estimate (RMSE) | Models | Manta | Lumpy | GRIDSS | SvABA | Delly |
|---|---|---|---|---|---|---|
| Sensitivity | (1) | 0.041 | 0.034 | 0.051 | 0.052 | 0.021 |
| | (2) | **0.041** | 0.034 | 0.051 | 0.053 | 0.021 |
| | (3) | 0.042 | 0.034 | 0.051 | **0.052** | 0.021 |
| | (4) | 0.042 | 0.034 | 0.051 | 0.053 | 0.021 |
| | (5) | 0.042 | **0.034** | 0.051 | 0.053 | 0.021 |
| | (6) | 0.042 | 0.036 | **0.051** | 0.058 | **0.021** |
| | (7) | 0.068 | 0.061 | 0.092 | 0.091 | 0.026 |
| | (8) | 0.136 | 0.094 | 0.144 | 0.148 | 0.167 |
| Precision | (1) | 0.012 | 0.012 | 0.007 | 0.029 | 0.061 |
| | (2) | 0.012 | **0.012** | 0.007 | 0.029 | **0.061** |
| | (3) | **0.012** | 0.013 | 0.007 | 0.029 | 0.156 |
| | (4) | 0.013 | 0.013 | 0.007 | 0.029 | 0.156 |
| | (5) | 0.017 | 0.013 | **0.007** | **0.029** | 0.156 |
| | (6) | 0.019 | 0.017 | 0.007 | 0.032 | 0.156 |
| | (7) | 0.018 | 0.015 | 0.012 | 0.046 | 0.151 |
| | (8) | 0.024 | 0.020 | 0.013 | 0.055 | 0.148 |
| F1 score | (1) | 0.039 | 0.032 | 0.048 | 0.057 | 0.019 |
| | (2) | **0.039** | 0.032 | 0.048 | 0.058 | 0.019 |
| | (3) | 0.040 | 0.032 | 0.048 | **0.057** | 0.019 |
| | (4) | 0.040 | 0.032 | 0.048 | 0.058 | 0.019 |
| | (5) | 0.040 | **0.032** | 0.048 | 0.060 | **0.019** |
| | (6) | 0.040 | 0.034 | **0.048** | 0.065 | 0.020 |
| | (7) | 0.066 | 0.060 | 0.091 | 0.101 | 0.021 |
| | (8) | 0.132 | 0.088 | 0.143 | 0.163 | 0.146 |

## Web-based application design

Shiny-SoSV is a web application developed using R package shiny v1.3.2, and is hosted on shinyapps.io, built to provide a visual platform for exploring the behaviour of sensitivity and precision with various predictor variables through two main interactive plots. R objects of class "gam" from the GAM models make up the data underlying Shiny-SoSV. Manipulation of response variables by users via the web app are registered by R/shiny, which then signals a call to the function predict(), returning predicted sensitivity, precision and F1 score values along with estimated standard errors. Plots of predicted sensitivity and precision with confidence intervals are generated with R package ggplot2 version 3.1.1 [21]. Comparison and interaction effects of two or more predictors are possible by selecting additional variables, including selection of multiple SV callers via checkboxes and one or more numeric parameters via slider bars or checkboxes (Fig 5).

## Results and discussion

### Prediction model validation

To test the robustness and predictive ability of the models, we used somatic SV calling evaluation results from (1) an independent simulation set, (2) the ICGC-TCGA DREAM Somatic Mutation Calling Challenge (ICGC-TCGA DREAM Challenge) [11] and *in silico* mixing of cancer cell lines [22].

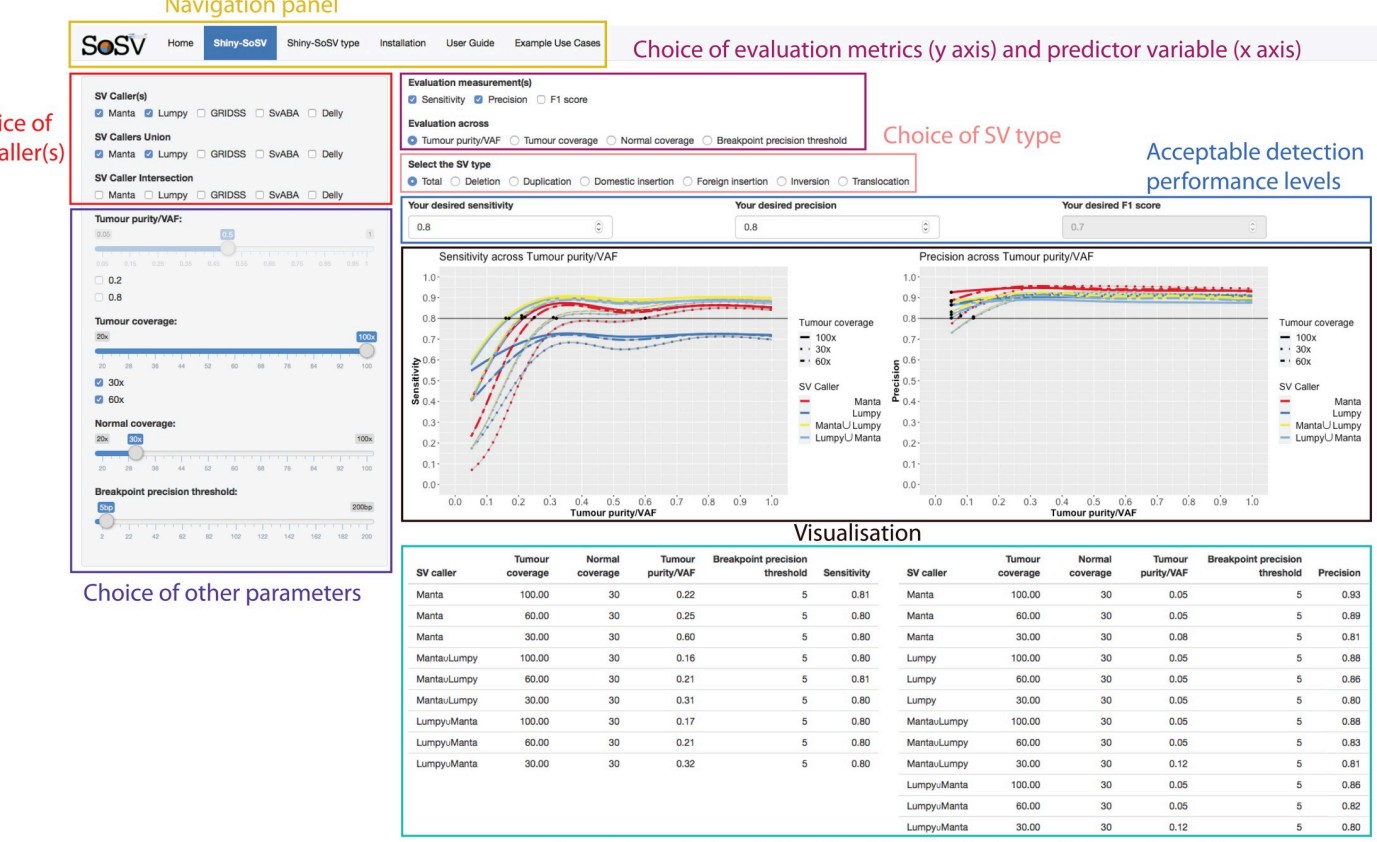

**Fig 5. Shiny-SoSV user interface.** Shown is a snapshot of Shiny-SoSV, displaying sensitivity and precision estimates evaluated across variant allele frequency (Tumour purity/VAF) (Black box labelled "Visualisation"). In this example, the overall performances of two SV callers (Manta and Lumpy) and their union callsets (distinguished by line colours) are displayed (controlled by red box labelled "Choice of SV caller(s)") for three different tumour depths of coverage settings (line types; 30x, 60x, 100x) (controlled by indigo coloured box labelled "Choice of other parameters"). The desired sensitivity and precision are set as 80% (Blue box labelled "Acceptable detection performance levels"). Confidence intervals of all estimates are shown as grey ribbons, which may be difficult to see due to tight intervals.

**Validation on an independent simulation set.** An independent simulation set was generated for validation using a separate simulated SV set (S1 Table) with exhaustive combinations of the following parameter values:

- Normal sample coverage: 25x, 45x, 60x, 75x, 90x;

- Tumour sample coverage: 25x, 45x, 60x, 75x, 90x;

- VAF: 0.10, 0.35, 0.50, 0.75, 0.90;

- Breakpoint precision threshold (bp): 5, 40, 100, 150, 180.

The results in Table 2 suggest that we can predict sensitivity, precision and F1 score to within 6% error rate. Compared with the cross-validation error estimation for the chosen

**Table 2. Predictive model validation.**

| Error Estimate (RMSE) | Manta | Lumpy | GRIDSS | SvABA | Delly |
|---|---|---|---|---|---|
| Sensitivity | 0.044 | 0.045 | 0.053 | 0.048 | 0.021 |
| Precision | 0.015 | 0.020 | 0.013 | 0.016 | 0.041 |
| F1 score | 0.044 | 0.042 | 0.051 | 0.055 | 0.015 |

prediction models in Table 1, all predictions with this independent data set achieved similar error rate (difference within 1%), with the exception of precision estimates for SvABA (1.3% lower RMSE) and Delly (2% lower RMSE). Relative to other callers, SvABA and Delly report fewer (< 10) total number of detected SVs at the combinations of low tumour coverage (20x), VAF (0.05) and breakpoint precision threshold (2bp), resulting in imprecise precision estimation, which has the effect of inflating error rate. As the lowest tumour coverage, VAF and breakpoint precision threshold simulated for this independent validation data set are higher than the range where SvABA and Delly struggle to make positive calls, it resulted in a lower error rate estimates for precision prediction for these two callers.

**Validation using evaluation results from synthetic data using cell lines.** The ICGC-TCGA DREAM Challenge conducted a crowdsourced benchmarking of SV callers and reported results on three different simulated tumours from 15 teams [11]. Each tumour sample was simulated with SVs spiked into a sub-sampling of Illumina paired-end short-read sequencing reads from a cell line using BAMSurgeon [11] and each matched-normal sample was derived from a separate (non-overlapping) sub-sampling of the same original BAM file. Both of tumour and matched normal samples were sampled to around 30-40x coverage. Three synthetic datasets (denoted as in_silico 1–3) were generated, based on two different cell lines and three different SV sets containing different types and numbers of SVs (371, 655 and 2,886) spiked in at different frequencies (100%, 80% and mixture of subsets of SV at 20%, 33%, and 50% VAF). Detailed description of the challenge and datasets can be found at https://www.synapse.org/#!Synapse:syn312572/wiki/62018. SVs reported by each team were evaluated based on a single criterion that the called SV breakpoint is within 100 bp of a simulated SV, while correct SV type was not required. So, this TP criterion is more lenient than the TP criteria used in our study. Among the teams that participated, Teams 1 and 2 used Delly and Manta respectively, which allowed their sensitivity and precision to be predicted using our GAM models and compared to their reported evaluation results (Fig 6). Teams were allowed to use different versions and parameters on different datasets and make multiple submissions for each dataset, which resulted in multiple sensitivity and precision results for each team and dataset combination (e.g. sensitivity ranging from 53% to 74% of Team 1 for in_sillico 3). Overall, our models achieved low error rate (MAE = 3.4%) for precision but higher error rate (MAE = 9.3%) for sensitivity prediction, especially for in_silico 2 and 3. This is likely due to a combinations of differences in simulation and evaluation methods, and different versions and parameters of SV callers used by the DREAM Challenge participants [11].

**Validation using evaluation results from in silico mixing of cancer cell lines.** We also validated our prediction models using somatic SV evaluation results based on two cancer cell lines (COLO-829 and HCC-1143), recently reported by Arora *et al.* [22]. In that study, the authors performed whole-genome sequencing of the cancer cell lines along with their matched normal cell lines, to up to 278x coverage. By down-sampling and mixing different fractions of data from the tumour and normal samples, the authors evaluated somatic SV calling performance for different tumour purity/VAF (at 80x tumour and 40x normal coverage) and different tumour and normal coverages (at VAF of 1). This allowed us to compare our prediction curve to their reported SV performance on cancer cell lines with predictor variables VAF (Fig 7), tumour coverage (Fig 8) and matched normal coverage (Fig 9).

In the study of Arora *et al.*, evaluation results were reported for two callsets: AllSomatic and HighConf. The AllSomatic set was defined as the union set of three SV callers (SvABA, Manta and Lumpy) and filtering on a panel of normal, while the HighConf set is a subset of AllSomatic set, including SVs called by two of the three callers or called by Manta or Lumpy with either additional CNV or split-read support [22]. For validation, we compared Arora *et al.*'s results with our prediction curves derived from our union and intersection callsets for all

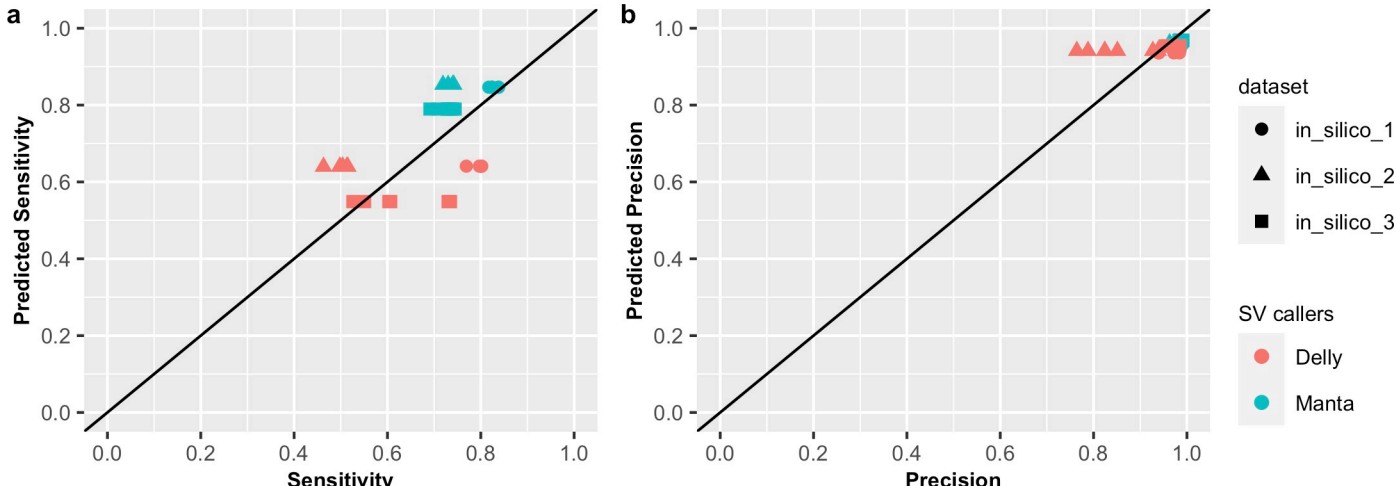

**Fig 6. Comparison of GAM prediction and observed performance from the ICGC-TCGA DREAM Challenge.** Shown are predicted sensitivity (a) and precision (b) by GAM model versus reported sensitivity and precision from Lee et al. [11]. Team 1 and Team 2 have used SV callers of Delly and Manta respectively. Sensitivity and precision are predicted based on predictor variables of tumour and normal coverage as 30x, breakpoint precision threshold as 100 bp and VAF as 100%, 80% and 50% for three datasets respectively.

pairwise combination of the three SV callers. Overall, our prediction model of these three variables (VAF, tumour and normal coverages) on SV detection is similar to the evaluation results reported by Arora *et al*. Specifically, we verified that VAF has a non-linear effect on sensitivity, and VAF and tumour coverage have stronger impacts on sensitivity than precision. Interestingly, we note that the evaluation results reported by Arora *et al*. is more similar to our predictions for the intersection callset for Manta and Lumpy and union callset for Lumpy and SvABA. This suggests these latter two callsets perform similarly to the additional filtering steps employed by Arora *et al*. Furthermore, we note that, in general, our prediction of sensitivity for pairwise intersection callsets are lower than reported by Arora *et al*. We believe this is likely due to our more stringent definition for concordance of two callsets, requiring breakpoint precision threshold of 5bp, compared to Arora et al.'s requirement of 300bp and at least 50% reciprocal overlap.

## Application features

The aim of Shiny-SoSV is to support decision-making on variables impacting one's ability to detect somatic SVs from WGS data. In particular, this app was designed to empower users in planning their WGS experiment for efficient somatic SV detection. To this end, seven key features were implemented.

- *Inclusion of important predictor variables*. We have included five variables in Shiny-SoSV. With the exception of VAF, all can be modified by the study investigator prior to the commencement of sequencing. This is important as it allows users to adjust experimental plans accordingly. Even though VAF is intrinsic to the biology of the sample, it is by far the most important predictor for somatic variant detection [11, 23]. Therefore, VAF is included mainly to determine minimum required tumour purity and support tuning of other modifiable variables.

- *Predictor variables are easy to modify*. To make it user friendly, we have implemented tick boxes and slider bars for value selection across a range of variable choices. For example, users can choose any single or pairs of SV caller(s).

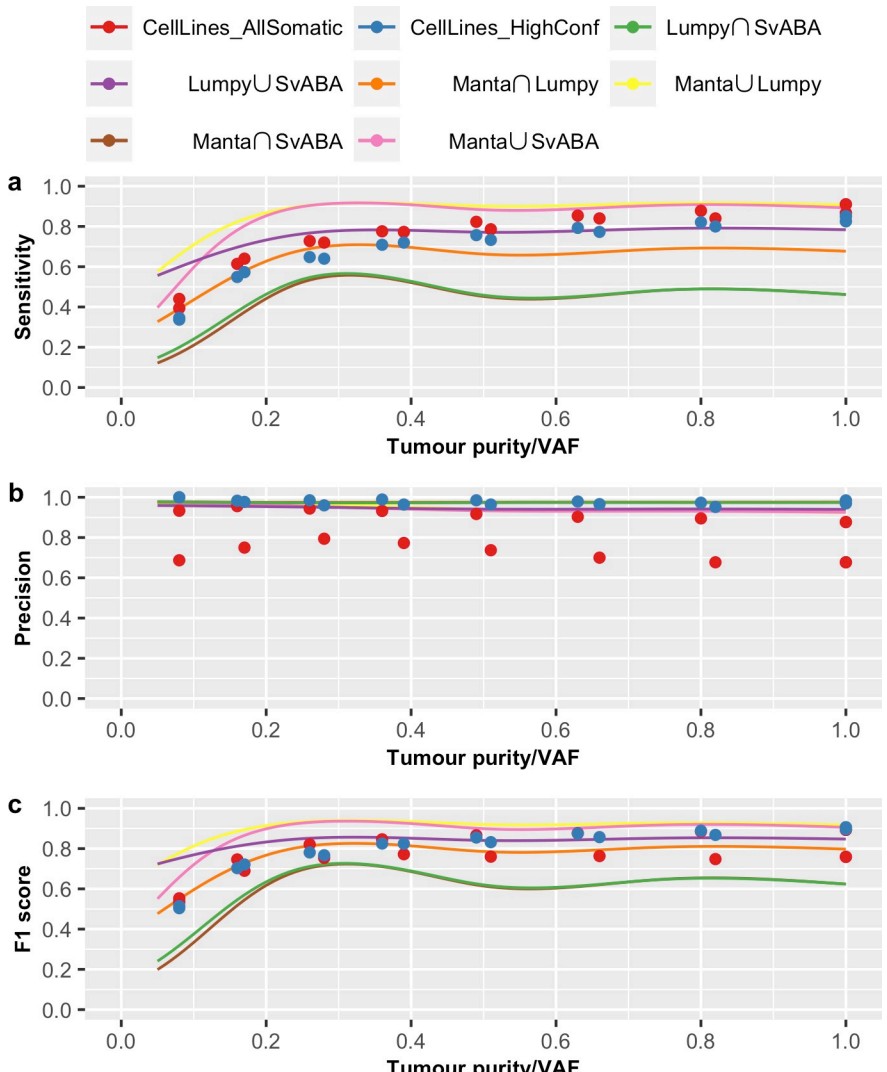

**Fig 7. Prediction curve across tumour purity/VAF.** Shown are the evaluation results of AllSomatic set (in red) and HighConf set (in blue) from cancer cell lines and prediction curve of (a) sensitivity, (b) precision and (c) F1 score across tumour purity/VAF. Prediction are based on tumour purity/VAF from 0.05 to 1, tumour coverage of 80x, normal coverage of 40x and breakpoint precision threshold of 200bp.

- *Intuitive visualisation of the impact of predictor variables on responses.* To compare somatic SV detection performance with different choices of parameter values, GAM prediction models were implemented allowing prediction curves to be dynamically generated for easy visualisation. The prediction can be visualised via line plots with confidence intervals based on choices of different combinations of variables, in one figure.

- *Ability to evaluate multiple variables simultaneously.* For example, users can examine the change of sensitivity (y-axis) across VAF (x-axis) with different choices of SV callers (line colour) and tumour coverages (line types) (Fig 4).

- *Simple reporting of minimum requirements to achieve user's objective.* To further simplify decision-making, a summary table of minimum required value for each predictor variable,

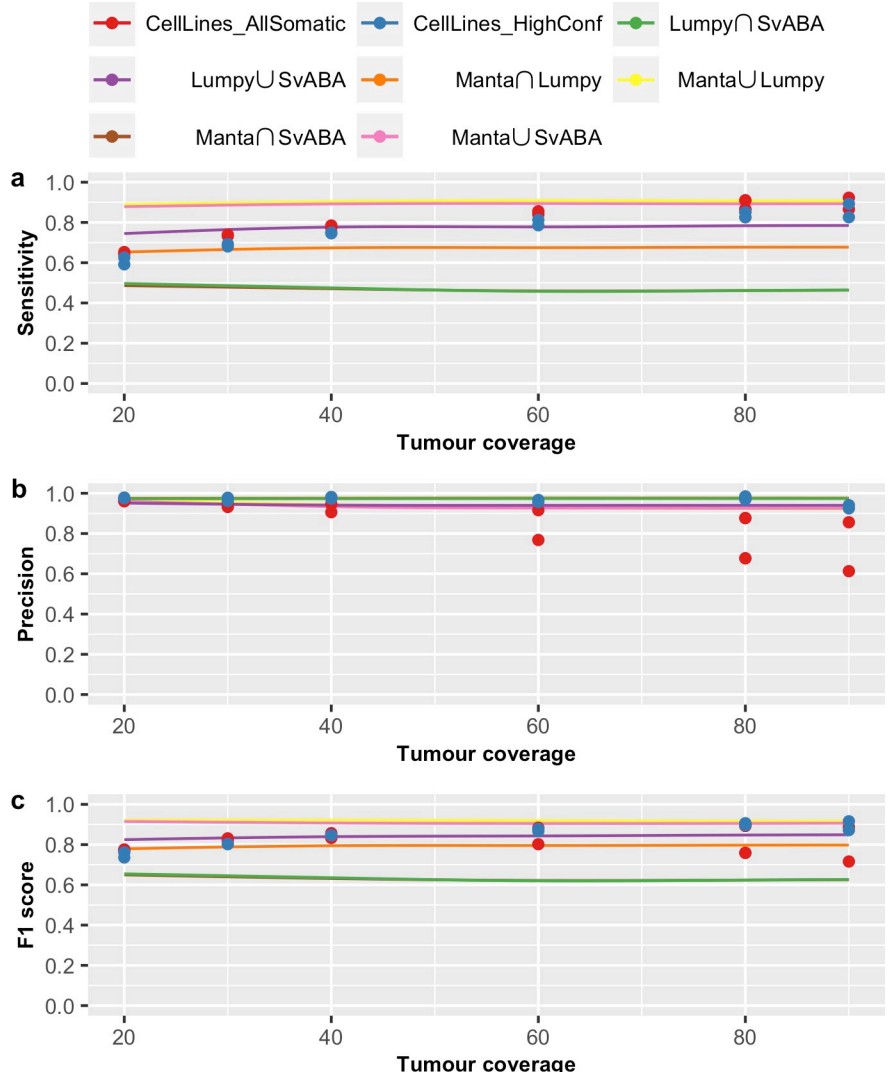

**Fig 8. Prediction curve across tumour coverage.** Shown are the evaluation results of AllSomatic set (in red) and HighConf set (in blue) from cancer cell lines and prediction curve of (a) sensitivity, (b) precision and (c) F1 score across tumour coverage. Prediction are based on tumour coverage from 20x to 90x, tumour purity/VAF of 100%, normal coverage of 40x and breakpoint precision threshold of 200bp.

including SV callers, depth of coverage and tumour purity and breakpoint resolution, is reported for a user-defined detection performance level.

- *Inclusion of commonly used SV callers, representing different SV calling methods.* The five SV callers included in Shiny-SoSV are commonly used and have shown good performance in multiple recent benchmarking studies. They were also chosen to represent different combinations of SV calling methods. The simulated samples and SV set can be used for evaluation of future callers and be added to Shiny-SoSV.

- *Ability to estimate performance by SV type.* In addition to the main user interface for prediction of overall SV detection performance, users can further predict and compare the performance across different SV types.

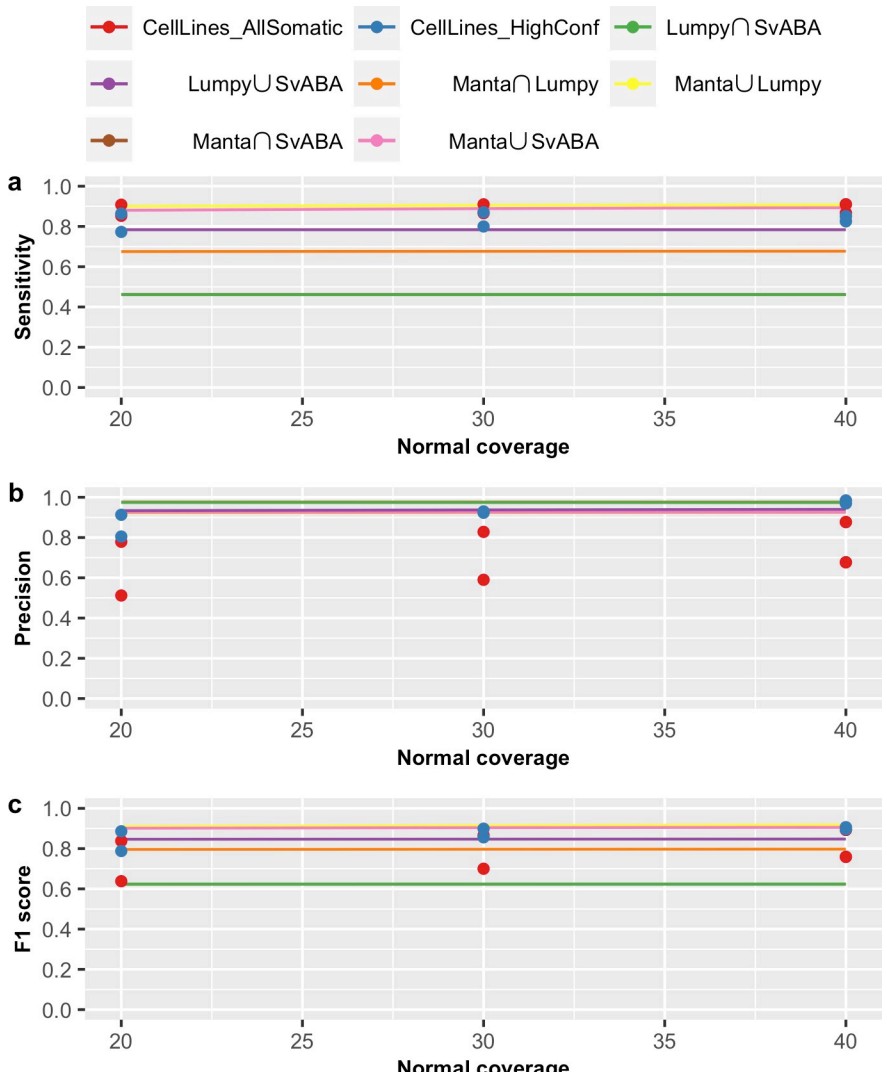

**Fig 9. Prediction curve across normal coverage.** Shown are the evaluation results of AllSomatic set (in red) and HighConf set (in blue) from cancer cell lines and prediction curve of (a) sensitivity, (b) precision and (c) F1 score across normal coverage. Prediction are based on normal coverage from 20x to 40x, tumour purity/VAF of 100%, tumour coverage of 80x and breakpoint precision threshold of 200bp.

## Application limitations

While we have endeavoured to create a performance calculator for real-world use, we recognise certain limitations and challenges. Firstly, the performance prediction was based on a series of simulated data, which we appreciate cannot fully encompass the complexity of real tumour sample and errors in real sequencing processes. However, by concurrently including both germline and somatic SVs, and including a range of SV types and SV sizes, we believe the simulation results provide a baseline estimate. Secondly, in general, real unknown VAF can be lower than histopathological estimates of tumour purity, due to potential tumour sub-clonality and heterogeneity. We therefore present the prediction here to represent the upper limits of sensitivity and precision that can be achieved. Thirdly, we have included four predictor variables totalling 10,692 unique permutations of predictor values for prediction model fitting of each callset. However, other variables, such as average mapping quality, read length, insert

sizes and nucleotide complexity, have been shown to impact variant detection, albeit with lower effect size [9, 11]. Adding more variables and more data points could provide a better prediction model and more sophisticated calculator for SV detection performance prediction.

## Application utility

The web-based app is hosted indefinitely at https://hcpcg.shinyapps.io/shiny-sosv by Shiny from RStudio. Both the input data to the GAM and R script for GAM fitting, as well as the resulting GAM R objects are provided with the Git release, available at https://github.com/tgong1/Shiny-SoSV. Additionally, the source code for launching the shiny app is available at this GitHub repository, allowing both data and app versions to be tracked. An extensive User Guide is provided with Shiny-SoSV release, detailing the user-interface and available options. To help further enhance usability, multiple end-user cases also accompany Shiny-SoSV.

To illustrate the utility of Shiny-SoSV here, we present a common example use-case.

*Suppose, based on histopathology, a user is aware their cohort of cancer samples have tumour purities between 20% and 60%, and they want to know to what depth of coverage they should sequence their tumour samples, assuming all matched normal samples will be sequenced to the default of 30x.*

As demonstrated in Fig 5, the Shiny-SoSV user can select to evaluate the effect of "Tumour purity/VAF" ("Evaluation across") on either or both "Sensitivity" and "Precision" ("evaluation measurements"). On the side bar, the user can select individual or combination(s) of SV callers (e.g. Manta, Lumpy, and their union set) for comparison and up to three tumour coverage settings (e.g. 30x, 60x and 100x) simultaneously, while keeping all other parameters unchanged. From this, it is immediately obvious that VAF has a great impact on sensitivity, while little impact on precision. Sensitivity of all SV callers increases rapidly from VAF of 5% to 30%. At VAF > 30%, improvements in sensitivity notably slows for all SV callers, with Manta showing relatively larger improvements until it reaches the limit (saturation) with this combination of parameters for this caller. The table below the plots provides the user with the information required to make decisions on sequencing depth, which is further dependent on his/her budget and objective (i.e. acceptable sensitivity and precision level). For example, for a desired sensitivity and precision of 80%, the user may elect to sequence at 60x using the SV caller(s) chosen (e.g. union set of Manta and Lumpy) for all samples. However, if the resulting cost is prohibitive, the user may choose to sequence only the subset of samples with tumour purity > 32% to up to 30x depth of coverage whilst still achieving the desired 80% sensitivity and precision levels.

## Conclusions

Shiny-SoSV provides an easy to use and interactive graphical user interface for evaluating the effects of multiple variables impacting somatic SV detection. The current release allows evaluation of common effectors of SV detection using five popular SV callers. Inclusion of addition SV callers can easily be incorporated with existing simulation datasets, while assessment of additional variables (such as mapping quality, insert sizes and nucleotide complexity) can be achieved with further simulation efforts. In sum, we believe Shiny-SoSV will enable bioinformaticians, and importantly also non-bioinformaticians, to optimally design WGS experiments for detecting SVs in cancer genomes.

## Supporting information

**S1 Fig. The joint impact of tumour purity/VAF and breakpoint precision threshold on sensitivity and precision.** Shown are the effect of the interaction of VAF and breakpoint precision

threshold (by colour) on sensitivity (a) and precision in log scale (b) for five SV callers (Manta, Lumpy, GRIDSS, SvABA, Delly). Results are based on simulation data set with tumour coverage of 20x, 60x and 90x, breakpoint precision threshold of 2bp, 60bp, 100bp and 200bp and normal coverage of 60x.
(TIF)

**S2 Fig. The joint impact of tumour coverage and tumour purity/VAF on sensitivity and precision.** Shown are the effect of the interaction of tumour coverage and VAF (by colour) on sensitivity (a) and precision in log scale (b) for five SV callers (Manta, Lumpy, GRIDSS, SvABA, Delly). Results are based on simulation data set with breakpoint precision threshold of 10bp, 100bp and 200bp, VAF of 0.05, 0.3 0.5 and 1 and normal coverage of 60x.
(TIF)

**S3 Fig. The joint impact of breakpoint precision threshold and tumour coverage on sensitivity and precision.** Shown are the effect of the interaction of breakpoint precision threshold and tumour coverage (by colour) on sensitivity (a) and precision in log scale (b) for five SV callers (Manta, Lumpy, GRIDSS, SvABA, Delly). Results are based on simulation data set with VAF of 0.2, 0.5 and 1, tumour coverage of 20x, 40x, 60x and 90x and normal coverage of 60x.
(TIF)

**S4 Fig. The joint impact of normal coverage and VAF on sensitivity and precision.** Shown are the effect of the interaction of normal coverage and VAF (by colour) on sensitivity (a) and precision in log scale (b) for five SV callers (Manta, Lumpy, GRIDSS, SvABA, Delly). Results are based on simulation data set with breakpoint precision threshold of 10bp, 100bp and 200bp, VAF of 0.05, 0.3 0.5 and 1 and tumour coverage of 60x.
(TIF)

**S5 Fig. Relationship between SV type sensitivity and tumour purity/VAF.** Shown are the effects of tumour purity/VAF on somatic SV type detection sensitivity for five SV callers (Manta, Lumpy, GRIDSS, SvABA, Delly).
(TIF)

**S6 Fig. Relationship between SV type sensitivity and tumour coverage.** Shown are the effects of tumour coverage on somatic SV type detection sensitivity for five SV callers (Manta, Lumpy, GRIDSS, SvABA, Delly).
(TIF)

**S7 Fig. Relationship between SV type sensitivity and normal coverage.** Shown are the effects of normal coverage on somatic SV type detection sensitivity for five SV callers (Manta, Lumpy, GRIDSS, SvABA, Delly).
(TIF)

**S8 Fig. Relationship between SV type sensitivity and breakpoint precision threshold.** Shown are the effects of breakpoint precision threshold on somatic SV type detection sensitivity for five SV callers (Manta, Lumpy, GRIDSS, SvABA, Delly).
(TIF)

**S9 Fig. Relationship between SV type precision and tumour purity/VAF.** Shown are the effects of tumour purity/VAF on somatic SV type detection precision for five SV callers (Manta, Lumpy, GRIDSS, SvABA, Delly). INS is not detectable by Lumpy. Precision of DUP for SvABA shown can also be INS.
(TIF)

**S10 Fig. Relationship between SV type precision and tumour coverage.** Shown are the effects of tumour coverage on somatic SV type detection precision for five SV callers (Manta, Lumpy, GRIDSS, SvABA, Delly). INS is not detectable by Lumpy. Precision of DUP for SvABA shown can also be INS.
(TIF)

**S11 Fig. Relationship between SV type precision and normal coverage.** Shown are the effects of normal coverage on somatic SV type detection precision for five SV callers (Manta, Lumpy, GRIDSS, SvABA, Delly). INS is not detectable by Lumpy. Precision of DUP for SvABA shown can also be INS.
(TIF)

**S12 Fig. Relationship between SV type precision and breakpoint precision threshold.** Shown are the effects of breakpoint precision threshold on somatic SV type detection precision for five SV callers (Manta, Lumpy, GRIDSS, SvABA, Delly). INS is not detectable by Lumpy. Precision of DUP for SvABA shown can also be INS.
(TIF)

**S13 Fig. Relationship between SV type F1 score and tumour purity/VAF.** Shown are the effects of tumour purity/VAF on somatic SV type detection F1 score for five SV callers (Manta, Lumpy, GRIDSS, SvABA, Delly). INS is not detectable by Lumpy. F1 score of DUP for SvABA shown can also be INS.
(TIF)

**S14 Fig. Relationship between SV type F1 score and tumour coverage.** Shown are the effects of tumour coverage on somatic SV type detection F1 score for five SV callers (Manta, Lumpy, GRIDSS, SvABA, Delly). INS is not detectable by Lumpy. F1 score of DUP for SvABA shown can also be INS.
(TIF)

**S15 Fig. Relationship between SV type F1 score and normal coverage.** Shown are the effects of normal coverage on somatic SV type detection F1 score for five SV callers (Manta, Lumpy, GRIDSS, SvABA, Delly). INS is not detectable by Lumpy. F1 score of DUP for SvABA shown can also be INS.
(TIF)

**S16 Fig. Relationship between SV type F1 score and breakpoint precision threshold.** Shown are the effects of breakpoint precision threshold on somatic SV type detection F1 score for five SV callers (Manta, Lumpy, GRIDSS, SvABA, Delly). INS is not detectable by Lumpy. F1 score of DUP for SvABA shown can also be INS.
(TIF)

**S1 Table. Simulation SV sets.**
(XLSX)

**S2 Table. Predictive model comparison and selection based on MAE.**
(DOCX)

**S3 Table. The parametric coefficients and approximate significance of smooth terms of the selected GAM models.**
(DOCX)

**S4 Table. Predictive model comparison and selection for each SV type.**
(DOCX)

**S1 File.**
(DOCX)

## Acknowledgments

We gratefully acknowledge the comments and suggestions on the manuscript from Dr Timothy Peters at Garvan Institute of Medical Research.

We thank Nicolas Robine, Minita Shah and Jennifer Shelton from the New York Genome Center for providing the evaluation data for cell lines COLO829 and HCC-1143.

We acknowledge the high-performance computing resources generously provided by the National Computational Infrastructure (Raijin), the Garvan Institute of Medical Research (Wolfpack) and the University of Sydney (Artemis).

## Author Contributions

**Conceptualization:** Tingting Gong, Eva K. F. Chan.

**Data curation:** Tingting Gong.

**Formal analysis:** Tingting Gong.

**Funding acquisition:** Vanessa M. Hayes.

**Investigation:** Tingting Gong, Eva K. F. Chan.

**Methodology:** Tingting Gong, Eva K. F. Chan.

**Project administration:** Eva K. F. Chan.

**Resources:** Vanessa M. Hayes.

**Software:** Tingting Gong.

**Supervision:** Vanessa M. Hayes, Eva K. F. Chan.

**Validation:** Tingting Gong.

**Visualization:** Tingting Gong.

**Writing – original draft:** Tingting Gong.

**Writing – review & editing:** Tingting Gong, Vanessa M. Hayes, Eva K. F. Chan.

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
