## [Decision Letter · Decision Letter 0]

8 Jun 2020

PONE-D-20-12646

Shiny-SoSV: A web-based performance calculator for somatic structural variant detection

PLOS ONE

Dear Dr. Chan,

Thank you for submitting your manuscript to PLOS ONE. After careful consideration, we feel that it has merit but does not fully meet PLOS ONE’s publication criteria as it currently stands. Therefore, we invite you to submit a revised version of the manuscript that addresses the points raised during the review process.

We look forward to receiving your revised manuscript.

Kind regards,

Zechen Chong

Academic Editor

PLOS ONE

Journal Requirements:

'T.G. is funded by an Australian Government Research Training Program Scholarship, E.K.F.C

is funded in part by a Movember Revolutionary Team Award (PRoMis) from Movember

Australia and the Prostate Cancer Foundation Australia, and V.M.H. is funded by the

University of Sydney Foundation and Petre Foundation, Australia.'

'The funders had no role in study design, data collection and analysis, decision to

publish, or preparation of the manuscript.'

Additional Editor Comments (if provided):

Reviewers' comments:

Reviewer's Responses to Questions

**Comments to the Author**

1. Is the manuscript technically sound, and do the data support the conclusions?

Reviewer #1: Yes

Reviewer #2: Yes

Reviewer #3: Yes

2. Has the statistical analysis been performed appropriately and rigorously? 

Reviewer #1: Yes

Reviewer #2: Yes

Reviewer #3: Yes

3. Have the authors made all data underlying the findings in their manuscript fully available?

Reviewer #1: Yes

Reviewer #2: Yes

Reviewer #3: Yes

4. Is the manuscript presented in an intelligible fashion and written in standard English?

Reviewer #1: Yes

Reviewer #2: Yes

Reviewer #3: Yes

5. Review Comments to the Author

Reviewer #1: The authors developed a web-based interactive app to help evaluate the effects of multiple variables on somatic SV detection sensitivity and precision. The impact of sequencing coverage, tumor purity, breakpoint precision threshold and performance of 5 SV caller tools were assessed through building generalised additive models. I think it contributes towards easy-use tools for researchers to plan their WGS experiment, especially for choosing desired sample coverage which affects the sequencing cost.

The manuscript is well-written and the webpage is nicely built. The app is easy to use with nice design and I like the descriptive and practical illustrations of examples albeit the prediction model is relatively simple with just four variables included as discussed in the limitation section.

I would like to see the following points to be addressed.

1. Since structural variations are generally defined into 5 categories: deletions (DELs), insertions (INSs), duplications (DUPs), inversions (INVs), and translocations (TRAs), and some SV callers performs better with certain SV types according to the information they leveraged to call variation, it would be helpful to see the prediction curves with each SV class besides the pooled sensitivity and precision if researchers have interest in a particular SV type. If there is a difference, I would be keen to see further discussion about whether it resulted from the underlying algorithm based on read depth, paired-end read discordance, or split-reads or other possible sources.

2. In the model validation section, both simulated dataset and cell-line synthetic datasets from ICGC-TCGA DREAM were used. Tumour cell line, however, is still simplification phenomena compared with real cancer tissues. I wonder how the model performs with in silico mixtures of real tumour samples.

3. The authors used sensitivity and precision as evaluation metrics and built two prediction models respectively. F1 score is the harmonic mean of precision and sensitivity and a single prediction curve can help users to make choice more easily.

4. As for the selected GAM prediction models, what are the coefficients and significance of both the main and interaction effects of used predictors? A supplementary table may help to understand the variable effects.

Other than those there were the following minor observations:

5. p.7, first paragraph of “Prediction Model Validation”. The two “predict”s are redundant in “The results suggest that we can predict sensitivity and precision prediction”.

6. Figure legend of Fig. S4. Missing space in “simulation data Set 1with tumour coverage”.

7. p.9, second point in “Application limitations”. Does the real unknown VAF is always lower than histopathological estimates so the predictions here are the upper limits?

8. The data points in scatterplots of Fig.1 and Fig.2 are overlapped, making it hard to inspect different sets.

Reviewer #2: The author described a tool called the Shiny-SoSV, which provides an interactive and visual platform for users to compare individual and combined impact of different parameters in somatic structural variant detection.

The author used simulation data to help estimate the performance (sensitivity and precision) of

somatic SV detection. And three parameters were included in the Shiny-SoSV tool, which was: variant allele frequency (VAF); sequencing depth of tumor and matched-normal samples; SV breakpoint precision.

In this study, the author detected somatic SVs using five SV callers: Manta, Lumpy, GRIDSS, Delly and SvABA for each tumour/normal pair and high-confidence calls were post filtered as described previously. However, the author didn’t give reason for why they used these five SV callers. There are many other popular tools. It will be more helpful if the author could simply introduce the tools, and the reason for selecting these five SV callers.

Reviewer #3: This paper presented a web-based tool , Shiny-SoSV, for predicting performance of somatic SV calling with different tumor purity and sequencing depth. The predictive model was selected and trained with simulation datasets and was validated with another two in-silico datasets. Shiny-SoSV was able to accurately predict somatic SV detection performance for five commonly used SV callers. It can advise users on the selection of SV caller or sequencing depth. There are some concerns about this paper:

1.As mentioned in Application limitations, the predictive model was trained with simulated datasets, and validated using another simulation replicate and an in-silico dateset from ICGC-TCGA challenge. In real tumor/normal sample sequencing data, the situation can be way more complicated than simulation. The model can be more confident if some real datasets are included for validation.

2.Overall, the simulation experiment design was well explained. Only the part about simulation replicates (three replicates Set 1-3, and three replicates of Set 1 itself) was confusing, as well as Table 1. Table 1 can be modified to show how the replicates was designed more intuitively.

3.About the simulated SV set, there were 1,200 germline SVs and 1,200 somatic SVs for each simulation. But there are around 20,000-30,000 SVs in normal human genome. This much lower number of germline SVs may facilitate somatic SV detection in the simulation, as there were much less noisy reads containing germline SV breakpoints.

4.The results for model validation with Set3 in Table 3 are very interesting. For sensitivity, in four out of five SV callers , RMSE and MAE are lower than RMSE and MAE with same Model in Table 2. But for precision, all five SV callers have higher RMSE and MAE in Set3. More explanation or discussion are expected than just showing RMSE and MAE values in the table.

6. PLOS authors have the option to publish the peer review history of their article (what does this mean?). If published, this will include your full peer review and any attached files.

Reviewer #1: Yes: Cong Zhang

Reviewer #2: No

Reviewer #3: Yes: Yu Chen

---

## [Author Response · Author response to Decision Letter 0]

8 Aug 2020

Reviewer #1: The authors developed a web-based interactive app to help evaluate the effects of multiple variables on somatic SV detection sensitivity and precision. The impact of sequencing coverage, tumor purity, breakpoint precision threshold and performance of 5 SV caller tools were assessed through building generalised additive models. I think it contributes towards easy-use tools for researchers to plan their WGS experiment, especially for choosing desired sample coverage which affects the sequencing cost.

The manuscript is well-written and the webpage is nicely built. The app is easy to use with nice design and I like the descriptive and practical illustrations of examples albeit the prediction model is relatively simple with just four variables included as discussed in the limitation section.

I would like to see the following points to be addressed.

<< Authors’ Response to Reviewer 1 >> 

#RESPONSE: We thank Reviewer 1 for their positive and constructive comments. We address Reviewer 1’s points below: 

1. Since structural variations are generally defined into 5 categories: deletions (DELs), insertions (INSs), duplications (DUPs), inversions (INVs), and translocations (TRAs), and some SV callers performs better with certain SV types according to the information they leveraged to call variation, it would be helpful to see the prediction curves with each SV class besides the pooled sensitivity and precision if researchers have interest in a particular SV type. If there is a difference, I would be keen to see further discussion about whether it resulted from the underlying algorithm based on read depth, paired-end read discordance, or split-reads or other possible sources.

#RESPONSE: We agree that a breakdown of performance metrics by SV type would be valuable and have done this in our revised manuscript. Specifically, we have expanded the Materials and Methods section “Defining true positive calls and concordant callsets” and “Predictive model selection”, included a new section “Evaluation and predictive model selection of SV type” in supplementary material S1 File, as well as added an additional “tab” in Shiny-SoSV for visualisation and prediction of SV caller performance by SV type.

#RESPONSE: In general, the impact of the evaluated predictor variables on the detection performance of individual SV type is similar to the impact on overall (SV type aggregated) performance. In our revised manuscript, we examined eight candidate GAM models for each SV type and SV caller, and have chosen the model with lowest estimated RMSE, as summarized in S4 Table.

#RESPONSE: In brief, we did find a difference in performances for different SV types, where, in general, the best performance was for the detection of DELs for any combination of variables, while INS was hard to detect. These findings support our previous findings [10]. The variability in detection performance by different SV callers for different SV types is mainly due to differences in the callers’ underlying detection and filtering algorithms. In the case of FINS, inserted sequences are absent from the reference genome, which results in weak or an absence of read-alignment signature. Therefore, sensitivity for FINS detection is low and only small FINS, which are within library insert size (500bp), are detectable by the callers evaluated, with the exception of Manta. Programmatically, Manta reports large INS irrespective of whether the inserted sequences can be fully assembled. An interesting finding was general low sensitivity in DEL detection by SvABA, which is predominantly due to its high default cut-off (97bp) of SV sizes [10]. Finally, we showed that SV callers that integrate local-assembly (Manta, GRIDSS and SvABA) perform better in DUP detection than the other callers. 

2. In the model validation section, both simulated dataset and cell-line synthetic datasets from ICGC-TCGA DREAM were used. Tumour cell line, however, is still simplification phenomena compared with real cancer tissues. I wonder how the model performs with in silico mixtures of real tumour samples.

#RESPONSE: This is a very valid question. In order to address this, we have taken advantage of a recent study of cancer cell lines (COLO-829 and HCC-1143), where the authors examined somatic SV calling performance for different levels of tumour purity, tumour coverage and normal coverage [22]. We compared their evaluation results with our prediction models in a newly added section “Validation using evaluation results from in silico mixing of cancer cell lines” of our revised manuscript, including the addition of Figs 7-9. Although the study of Arora et al. [22] did not present SV calling performance for individual callers, their reported sensitivity, precision and F1 score of their AllSomatic and HighConf callsets could be compared with our predictions of pairwise union and intersection callsets. Overall, we found similar trends of predicted SV calling performance with their evaluation results.

3. The authors used sensitivity and precision as evaluation metrics and built two prediction models respectively. F1 score is the harmonic mean of precision and sensitivity and a single prediction curve can help users to make choice more easily.

#RESPONSE: We agree that the predictive curve of F1 score can help users make quick decisions aiming to balance the sensitivity and precision. A new Fig 3 has been added to show the relationship between F1 score and predictor variables for each SV caller. The candidate models (Model (1)-(8)) were expanded and generalized to include F1 score as a response variable in the revised manuscript. Consequently, cross-validation error estimations of candidate models for F1 score were also added to Table 1 and Table S2. Furthermore, the predictive curve of F1 score has now been added alongside sensitivity and precision response variables in the main user interface of the Shiny-SoSV app. The accompanying user’s guide as well as example use-cases have also been updated to demonstrate the utility of F1 score prediction curve.

4. As for the selected GAM prediction models, what are the coefficients and significance of both the main and interaction effects of used predictors? A supplementary table may help to understand the variable effects.

#RESPONSE: Thank you for a good suggestion. We have now added a supplementary table (Table S3) for the results of GAM prediction models, including the coefficients and significance of terms the models.

Other than those there were the following minor observations:

5. p.7, first paragraph of “Prediction Model Validation”. The two “predict”s are redundant in “The results suggest that we can predict sensitivity and precision prediction”.

#RESPONSE: The sentence has been modified to remove one “predict”. 

6. Figure legend of Fig. S4. Missing space in “simulation data Set 1with tumour coverage”.

#RESPONSE: Spaced added.

7. p.9, second point in “Application limitations”. Does the real unknown VAF is always lower than histopathological estimates so the predictions here are the upper limits?

#RESPONSE: In practice, VAF can be lower or higher than histopathological estimates. However, in our experience, the real unknown VAF is typically lower than histopathology estimates of tumour purity predominantly because of tumour heterogeneity (presence of clonal and sub-clonal tumour cells). VAF simulated in this study represent tumour purity, which is the fraction of tumour to normal cells. The intent of the stated sentence was to bring awareness to the fact that tumour heterogeneity causing lower VAF for subsets of SVs, which in turn can be a limitation for accurately predicting overall SV detection performance. 

#RESPONSE: Further compounding this complication are that, in practice, there is significant variability in histopathological estimates of tumour purity between pathologists and that the actual piece of tissue used for clinical pathology is almost always different to the tissue sample used for sequencing. All these factors can result in discordance between sequencing and biochemical estimates of VAF. However, this discussion is beyond the scope of this paper on our web-based application. 

8. The data points in scatterplots of Fig.1 and Fig.2 are overlapped, making it hard to inspect different sets.

#RESPONSE: Fig 1 and Fig 2 have been modified to only contain one dataset. In addition, point size has been reduced to make them easier to see among different variables. 

Reviewer #2: The author described a tool called the Shiny-SoSV, which provides an interactive and visual platform for users to compare individual and combined impact of different parameters in somatic structural variant detection. The author used simulation data to help estimate the performance (sensitivity and precision) of somatic SV detection. And three parameters were included in the Shiny-SoSV tool, which was: variant allele frequency (VAF); sequencing depth of tumor and matched-normal samples; SV breakpoint precision.

In this study, the author detected somatic SVs using five SV callers: Manta, Lumpy, GRIDSS, Delly and SvABA for each tumour/normal pair and high-confidence calls were post filtered as described previously. However, the author didn’t give reason for why they used these five SV callers. There are many other popular tools. It will be more helpful if the author could simply introduce the tools, and the reason for selecting these five SV callers.

<< Authors’ Response to Reviewer 2 >> 

#RESPONSE: We thank Reviewer 2 for raising this important point and we agree that a brief description of these tools and why we chose them would be helpful to readers who are not intimately involved with somatic SV detection. We have now provided a brief explanation in the Methods section “Somatic SV detection”, as quoted here:

#RESPONSE: “These five callers were chosen to provide a wide representation of different SV detection methods and because they have been shown to be best performers within their class in recent benchmarking studies [9-12]. Specifically, both Delly and Lumpy use discordant read-pair and split-read methods, but Lumpy integrate two methods into one SV detection step, called “evidence clustering”, whereas Delly use them in two separate calling and refining steps [10]. Manta, GRIDSS and SvABA further use local-assembly with different methods of targeted assembly, windowed local assembly and genome-wide break-end assembly respectively [10]. Additionally, Manta, Lumpy, GRIDSS and Delly were identified as popular tools among 46 callers published from 2009 to 2017, based on the criteria of Web of Science counts in a recent benchmarking study [12]. While SvABA was published in 2018 and not included in that study, it has shown good performance for both germline and somatic SV detection in other studies [9, 10].”

#RESPONSE: In anticipation of new and better SV callers being developed in the future, we have made the simulated BAM files and simulated SV sets available via GitHub, with the idea that additional SV callers can be evaluated under identical conditions and so can be compared with the callers already available in Shiny-SoSV.

 

Reviewer #3: This paper presented a web-based tool , Shiny-SoSV, for predicting performance of somatic SV calling with different tumor purity and sequencing depth. The predictive model was selected and trained with simulation datasets and was validated with another two in-silico datasets. Shiny-SoSV was able to accurately predict somatic SV detection performance for five commonly used SV callers. It can advise users on the selection of SV caller or sequencing depth. There are some concerns about this paper:

<< Authors’ Response to Reviewer 3>> 

#RESPONSE: We thank Reviewer 3 for their thoughtful comments, which we address below: 

1. As mentioned in Application limitations, the predictive model was trained with simulated datasets, and validated using another simulation replicate and an in-silico dateset from ICGC-TCGA challenge. In real tumor/normal sample sequencing data, the situation can be way more complicated than simulation. The model can be more confident if some real datasets are included for validation.

RESPONSE: We appreciate this concern from the reviewer. Unfortunately, for obvious reasons, validation with real datasets is difficult because true positives and negatives are generally unknown. This was our original reasoning for using the ICGC-TCGA challenge dataset. However, recognising the need to demonstrate the validity of our prediction model with “more realistic” data, we have since used the published evaluation results of two cancer cell lines (COLO-829 and HCC-1143) for this purpose. In the study of Arora et al. [22], the authors examined somatic SV calling performance on cancer cell lines for different levels of tumour purity, tumour coverage and normal coverage. We compared their evaluation results with our prediction models in a newly added section “Validation using evaluation results from in silico mixing of cancer cell lines” of our revised manuscript, including the addition of Figs 7-9. Although the study of Arora et al. [22] did not present SV calling performance for individual callers, their reported sensitivity, precision and F1 score of their AllSomatic and HighConf callsets were compared with our predictions of pairwise union and intersection callsets. 

#RESPONSE: In summary, we found good concordance between our prediction model and all three validation datasets derived from different settings of increasing complexity. 

2. Overall, the simulation experiment design was well explained. Only the part about simulation replicates (three replicates Set 1-3, and three replicates of Set 1 itself) was confusing, as well as Table 1. Table 1 can be modified to show how the replicates was designed more intuitively.

#RESPONSE: We have now made the modification to describe the simulated and evaluated variables for model selection and validation separately under the sections of “Simulation of short paired-end reads” and “Predictive model validation” respectively in the main text, instead of in a table format, as following:

#RESPONSE: “The simulation was conducted to include a comprehensive combination of depths of coverage of the normal and tumour samples, VAF and SV breakpoint precision threshold (explained below in section “Defining true positive calls and concordant callsets”). The following parameter values were simulated:

Normal samples coverage: 20x, 30x, 40x, 50x, 60x, 70x, 80x, 90x, 100x

Tumour sample coverage: 20x, 30x, 40x, 50x, 60x, 70x, 80x, 90x, 100x

VAF: 0.05, 0.10, 0.20, 0.30, 0.40, 0.50, 0.60, 0.70, 0.80, 0.90, 1.00

Breakpoint precision threshold (bp): 2, 10, 20, 40, 60, 80, 100, 120, 140, 160, 180, 200

In all, this resulted in a total of 891 tumour/normal pairs of BAM files, encompassing all exhaustive combinations of parameter values.”

#RESPONSE: “An independent simulation set was generated for validation using a separate simulated SV set (S1 Table) with exhaustive combinations of the following parameter values:

Normal samples coverage: 25x, 45x, 60x, 75x, 90x;

Tumour sample coverage: 25x, 45x, 60x, 75x, 90x;

VAF: 0.10, 0.35, 0.50, 0.75, 0.90;

Breakpoint precision threshold (bp): 5, 40, 100, 150, 180.”

3. About the simulated SV set, there were 1,200 germline SVs and 1,200 somatic SVs for each simulation. But there are around 20,000-30,000 SVs in normal human genome. This much lower number of germline SVs may facilitate somatic SV detection in the simulation, as there were much less noisy reads containing germline SV breakpoints.

#RESPONSE: This is a very good point and we absolutely agree that in reality there are more germline SVs than somatic SVs in cancer genomes. Therefore, we have repeated our entire study by increasing the number of germline SVs by ten-fold, with new combinations of evaluated variables for both model selection and validation data sets. All results, including figures and tables, have been updated in the revised paper with these new results derived from the new simulation datasets.

#RESPONSE: We have further provided a detailed discussion of the impact of our simulation design to the SV evaluation results in section “SV simulation design comparison” of supplementary material S1 File. In brief, we observed no more than 5% change in sensitivity and less than 1% change in precision for most of the callers. The exceptions were the sensitivity of Manta and Delly, where we observed 2.5%-7.5% reduction for Manta and up to 15% for Delly at low VAF (< 0.5). 

4. The results for model validation with Set3 in Table 3 are very interesting. For sensitivity, in four out of five SV callers, RMSE and MAE are lower than RMSE and MAE with same Model in Table 2. But for precision, all five SV callers have higher RMSE and MAE in Set3. More explanation or discussion are expected than just showing RMSE and MAE values in the table.

#RESPONSE: We have added a discussion comparing error rate of validation data set in Table 2 and cross-validation error estimation for the chosen prediction models in Table 1 and an explanation on the change, especially the decreasing RMSE for precision prediction of SvABA and Delly.

---

## [Editor Report · Decision Letter 1]

11 Aug 2020

Shiny-SoSV: A web-based performance calculator for somatic structural variant detection

PONE-D-20-12646R1

Dear Dr. Chan,

We’re pleased to inform you that your manuscript has been judged scientifically suitable for publication and will be formally accepted for publication once it meets all outstanding technical requirements.

Kind regards,

Zechen Chong

Academic Editor

PLOS ONE
---

## [Editor Report · Acceptance letter]

18 Aug 2020

PONE-D-20-12646R1 

Shiny-SoSV: A web-based performance calculator for somatic structural variant detection 

Dear Dr. Chan:

I'm pleased to inform you that your manuscript has been deemed suitable for publication in PLOS ONE. Congratulations! Your manuscript is now with our production department. 

Kind regards, 

on behalf of

Dr. Zechen Chong 

Academic Editor

PLOS ONE